# Human Papilloma Virus Infection in Men: A Specific Human Virome or a Specific Pathology?

**DOI:** 10.3390/genes16020230

**Published:** 2025-02-18

**Authors:** Ivana Čulav, Mihael Skerlev, Lidija Žele Starčević, Pero Hrabač, Suzana Ljubojević Hadžavdić, Iva Bešlić, Liborija Lugović Mihić

**Affiliations:** 1Department of Dermatology, Children’s Hospital Zagreb, 10000 Zagreb, Croatia; 2Medical School Zagreb, St. Catherine’s Special Hospital, University of Zagreb, 10000 Zagreb, Croatia; skerlevm@gmail.com (M.S.); suzana.ljubojevic@gmail.com (S.L.H.); 3Department of Clinical and Molecular Microbiology, Medical School Zagreb, University Hospital Center Zagreb, University of Zagreb, 10000 Zagreb, Croatia; lidija.zele@gmail.com; 4Department of Medical Statistics, Epidemiology and Medical Informatics, Andrija Štampar School of Public Health, Medical School Zagreb, University of Zagreb, 10000 Zagreb, Croatia; pero.hrabac@mef.hr; 5Department of Dermatovenereology, University Hospital Center Sestre Milosrdnice, 10000 Zagreb, Croatia; ivaaabukvic@gmail.com; 6Department of Dermatovenereology, School of Dentistry Zagreb, University Hospital Center Sestre Milosrdnice, University of Zagreb, 10000 Zagreb, Croatia; liborija@gmail.com

**Keywords:** human papillomavirus (HPV), condylomata acuminata, high-risk HPV, microbiota, epidemiology, men’s health

## Abstract

Background: Human papillomavirus (HPV) infections in men remain under-researched despite their critical role in disease transmission and the increasing incidence of HPV-related cancers. This study investigates the clinical and molecular characteristics of anogenital HPV infections in men, emphasizing genotype prevalence, diagnostic methods, and lesion variability. Methods: A cross-sectional study was conducted on 70 men aged 18–65 years with clinically diagnosed anogenital HPV infection. Lesions were characterized by morphology and location. HPV DNA was analyzed using INNO-LiPA (INNOvative Line Probe Assay), Hybrid Capture II (HC II), and polymerase chain reaction (PCR) assays to determine genotype distribution. Associations between clinical features and HPV genotypes were assessed using multivariate statistical analyses. Results: Lesions varied in morphology, with verrucous (52.86%) and papular (30%) types being the most common. Localization patterns showed predominance on the penis radix (34.29%) and shaft (27.14%). Molecular testing revealed HPV DNA in 88.57% of the cases using INNO-LiPA, compared to 45% and 40% with HC II and PCR, respectively. Low-risk (LR) genotypes, particularly HPV6, dominated single infections, comprising 68.57% of the cases, while high-risk (HR) genotypes accounted for 20%. Mixed LR and HR infections were observed in 14.29% of the lesions, with greater diversity noted in distal genital regions. Notably, condyloma plana and lesions on the inner prepuce exhibited a higher prevalence of HR and mixed infections. Age and lesion duration showed trends toward older patients and longer disease duration in cases involving perianal and extragenital condylomas, though these findings were not statistically significant. No direct correlation between lesion type or localization and specific genotypes was identified, underscoring the heterogeneity of HPV clinical manifestations in men. Conclusions: Anogenital HPV infections in men exhibit significant heterogeneity in lesion morphology, localization, and genotype distribution. HR HPV genotypes were detected in a notable proportion of benign lesions, underscoring their potential role in disease progression. INNO-LiPA proved superior in diagnostic accuracy, highlighting the need for standardized and cost-effective diagnostic approaches for men. Further research is crucial to elucidate HPV’s clinical impact in men and inform prevention and treatment strategies.

## 1. Introduction

The human skin, the body’s largest organ with an estimated area of 25 m^2^, including appendages [1], serves as a critical barrier to the external environment through innate and adaptive immune functions [2]. Studies on the skin microbiome have highlighted the reciprocal interaction between host immunity and resident microbes; while host immunity shapes microbial composition, skin microbiota significantly influence immune mechanisms [3]. Dysregulation of this relationship can result in skin diseases [4]. A better understanding of microbial dynamics and ecology offers insights into skin disorders and novel therapeutic developments.

Compared to other organs like the intestine and lungs, the skin exhibits greater complexity due to diverse variables influencing its surface characteristics and unique cell types that interact with microbes [2,5]. Microbial diversity on the skin is determined by anatomical, environmental, and behavioral factors, leading to substantial inter- and intra-individual variability [6,7,8,9]. While most research focuses on bacterial species, viruses, fungi, and parasites also contribute to microbial diversity [2]. Viruses, despite their abundance and diversity, remain underexplored. The human virome includes approximately 10^13^ viral particles per individual, with much of its sequence data classified as “dark matter” [10,11].

Neonates are thought to lack a virome, with colonization beginning after delivery [12,13]. Viral colonization progresses through defined stages and exhibits significant inter-individual diversity [14]. The virome is relatively stable in healthy individuals but destabilizes during disease states [15]. Viruses in the skin virome include bacteriophages, viruses infecting microorganisms or human cells, and transient viruses from food [16]. Certain eukaryotic viruses, such as Adenoviridae, Herpesviridae, and Papillomaviridae, inhabit healthy skin, though their expansion is linked to immune deficiencies [17].

Papillomaviruses (HPVs) represent a diverse group of DNA viruses infecting humans and animals. They co-evolved with hosts over millions of years and often persist as commensals without causing disease [18]. Over 200 HPV genotypes are classified into five genera based on DNA sequence, with the majority belonging to the α and β genera [19]. Mucosal α HPVs are associated with carcinomas, while some β HPVs are linked to skin tumors [20]. Genotypes differ in oncogenic potential and clinical manifestations, influenced by host genotype, epithelial type, immunity, and environmental factors [21].

HPV infections typically spread through direct skin-to-skin or mucosa contact, often via microabrasions, though other routes include perinatal transmission and medical interventions [22]. HPV prevalence in asymptomatic men varies widely (1.3–72.9%) [23,24], with a large study reporting a prevalence of 65.2% in men aged 18–70 years [25,26]. Infections are frequently detected on the penis and surrounding areas, particularly in uncircumcised men [27]. Risk factors for HPV acquisition include sexual behavior, smoking, and partner characteristics, while protective factors include condom use and circumcision [28].

Most HPV infections regress spontaneously, particularly in younger individuals [29]. Clearance rates for low-risk and high-risk HPV are similar between men and women, with about 75% resolving within a year [30]. However, persistent high-risk HPV infections are linked to cancers, and recent years have seen rising incidences of HPV-related cancers in men, including penile, anal, and oropharyngeal carcinomas [31,32,33].

Anal carcinomas, primarily caused by HPV16 (87%) and HPV18 (9%), have increased by 2% annually over the past three decades [34]. Men who have sex with men (MSM), especially those engaging in receptive anal intercourse, face a significantly higher risk of anal carcinoma. Immunocompromised individuals and organ transplant recipients also exhibit higher risks [35]. Oropharyngeal carcinomas caused by HPV have risen, contrasting with declines in cases linked to smoking and alcohol. These HPV-related cancers display distinct biological behaviors and treatment responses [36].

Penile cancer, although rare, has shown an increasing incidence. About 50% of penile carcinomas are HPV-related, similar to HPV-associated vulvar cancers [21,33]. Risk factors include phimosis, chronic inflammation, poor hygiene, and smoking [32,37].

Specific therapies for anogenital HPV infections are unavailable, and current treatments often have prolonged and disappointing outcomes [38,39]. Preventive vaccines, including bivalent, quadrivalent, and nonavalent vaccines, target HPV before the onset of sexual activity [40]. Future strategies aim to develop curative vaccines and broaden coverage against more HPV genotypes, potentially reducing the global burden of HPV-related diseases and associated healthcare costs [41,42].

## 2. Patients and Methods

A cross-sectional study was conducted at a university hospital center outpatient clinic, involving 70 male patients aged 18–65 years diagnosed with anogenital HPV infection. Diagnosis was performed by two dermatovenereologists based on characteristic clinical manifestations and confirmed by HPV DNA typing. Clinically ambiguous cases were excluded. All participants provided informed consent, and the study protocol was approved by the institutional ethics committee.

After applying an anesthetic cream (EMLA^®^ (Aspen Pharma Tradind Ltd., Dublin, Ireland): 2.5% lidocaine and 2.5% prilocaine), two lesional skin or mucosa samples were collected from each patient using the excochleation technique. In cases where a larger single sample was collected, it was divided for analysis. A total of 140 samples were stored in Standard Transport Medium (STM) and transported to the laboratory on the same day. Each patient’s samples were analyzed using two molecular techniques to ensure robust results.

One sample per patient (*n* = 70) was used for HPV genotyping with INNO-LiPA, a reverse hybridization assay capable of identifying 32 HPV genotypes, including high- and low-risk types. The L1 region of the HPV genome was amplified using SPF_10_ primers, producing biotinylated amplicons. These were hybridized to specific oligonucleotide probes immobilized on membrane strips. Following stringent washing, streptavidin-conjugated alkaline phosphatase was applied, and a chromogenic reaction produced a visible purple precipitate, indicating genotype presence. Quality and DNA extraction were monitored using primers specific to the human HLA-DPB1 gene [43].

The second sample underwent HPV detection through either Hybrid Capture II (HC_2_) or an in-house PCR assay. HC_2_ is a nucleic acid hybridization assay with chemiluminescent signal amplification, targeting high-risk HPV DNA. RNA:DNA hybrids formed during the reaction were captured on antibody-coated microplates and detected using a luminometer. Results exceeding a predefined cutoff indicated the presence of high-risk HPV DNA [44].

The in-house PCR assay amplified a 450 bp fragment of the HPV L1 region using MY09/MY11 primers (Sigma-Aldrich, Saint Louis, MO, USA), which can detect over 50 HPV genotypes. β-globin gene amplification (268 bp) with GH20/PC04 primers served as an internal control, ensuring adequate DNA quality and extraction efficiency [45].

The findings of the two HPV detection methods were compared with INNO-LiPA genotyping to evaluate concordance and performance. Additional clinical data were collected, including condyloma morphology (papular, verrucous, fibromatoid, or plana), lesion localization (e.g., glans, sulcus coronarius, or perianal area), patient age, symptom duration, and HPV infection presence in sexual partners.

Multivariate statistical analyses examined the associations between clinical parameters and HPV genotypes. To ensure clarity despite the limited sample size, the results were grouped to minimize subgroup dispersion, enhancing data interpretability. This approach facilitated better visualization of correlations between clinical characteristics and HPV types. Statistical analysis was performed in Statistica software package v.14.0 [46] licensed to School of medicine, University of Zagreb.

## 3. Results

The clinical study involved 70 male participants aged 18 to 63 years, with a median age of 30.5 years. The duration of condylomata varied significantly, ranging from two weeks to 20 years, with a median duration of six months. The most common condyloma type observed was verrucous (52.86%), followed by papular (30%), while fibromatoid and plana types were less common (Table 1). The primary anatomical locations included the radix penis (34.29%) and the distal shaft (27.14%), with other localizations grouped for statistical clarity.

The molecular analysis of HPV DNA included the HC II assay, PCR, and genotyping using INNO-LiPA. Among 40 samples tested with the HC II assay, high-risk (HR) HPV DNA was detected in 45%. The PCR assay conducted on 50 samples revealed 40% positivity, while INNO-LiPA was positive in 62 of 70 (88.6%) samples. The INNO-LiPA method demonstrated its superiority in identifying HPV genotypes, detecting a broader range and higher frequency of positive results compared to HC II and PCR.

Genotyping results indicated that 70% of the 70 patients had one HPV genotype, while 18.58% had two or more. Eight samples (11.43%) were negative, predominantly associated with papular condylomas on the radix penis. Notably, HR HPV DNA was identified in 20% of the samples from benign lesions, with 14.29% showing mixed low-risk (LR) and HR HPV infections. This highlights the prevalence of HPV in these types of lesions and the diagnostic value of these methods. The exact prevalence of HPV genotypes is shown in Table 2.

Participants with condylomas on the inner prepuce exhibited a greater number of HPV genotypes compared to other localizations. Younger patients were more frequently associated with inner prepuce condylomas. This distribution underscores potential variations in HPV diversity based on anatomical site.

An alternative classification grouped condylomata into genital and extragenital regions, as well as proximal and distal genital regions. Genital lesions included those on the radix, shaft, and inner prepuce, while extragenital lesions were localized in the pubic, scrotal, and perianal areas. Distal genital lesions, such as those on the shaft and inner prepuce, showed a significantly higher number of HPV genotypes compared to proximal regions. This classification allows for a more nuanced understanding of the distribution of HPV genotypes in relation to lesion location.

Higher proportions of HR HPV genotypes were noted in condyloma plana compared to other types. Genital condylomas predominantly harbored LR genotypes, while HR genotypes were more common in distal regions, including the inner prepuce. Mixed LR and HR infections were observed across multiple condyloma types and locations. The proportion of genotypes according to risk grade is shown in Table 3. These findings indicate a complex interplay between HPV genotype risk levels and condyloma characteristics.

Age and duration analyses revealed that patients with perianal and extragenital condylomas tended to be older, with longer disease durations. However, these findings were not statistically significant. Nevertheless, they suggest a potential trend worth exploring in larger cohorts or with additional variables considered.

Among the sexual partners of participants, 28.57% were identified with an HPV infection. However, the presence of condylomata or positive HPV tests in partners did not significantly correlate with the type or localization of condylomas in participants. These findings underscore the need for improved diagnostic tools and further studies to better understand HPV transmission dynamics.

Further analyses explored the relationship between HPV genotypes and condyloma characteristics. Genital condylomas predominantly harbored LR genotypes, with HR genotypes being more prevalent in distal regions. Mixed LR and HR infections were observed across multiple types and localizations. These data emphasize the heterogeneity of HPV infections and their manifestations.

Alternative classifications by proximal and distal regions revealed that distal lesions exhibited significantly higher numbers of genotypes, particularly on the inner prepuce and shaft. This finding aligns with the understanding that certain anatomical sites may be more susceptible to diverse HPV infections due to variations in local immune responses or tissue characteristics. The median number of genotypes was 1.38 for distal regions compared to 0.95 for proximal regions (*p* = 0.032), reinforcing this distinction.

In summary, the study provides a comprehensive examination of the clinical and molecular features of condylomata and their association with HPV genotypes. By integrating data from multiple diagnostic methods and exploring alternative classification schemes, it highlights the complex interplay between HPV infections and clinical manifestations.

## 4. Discussion

Despite the abundant scientific activity on HPV, there is relatively limited investigation into its impact on the male population, particularly concerning male sexual health and the reproductive system. Men’s crucial role in disease transmission and the biological aspects of HPV infection emphasize the need for their inclusion in preventive strategies. Clinical studies with representative male samples are vital to understanding and addressing anogenital HPV infections. However, standardized diagnostic approaches for men with HPV are currently lacking, underlining the importance of accessible clinical and laboratory parameters for diagnosis and treatment planning. The lack of such standardization limits the ability to effectively implement strategies that could mitigate the impact of HPV on men and the broader community.

Our study investigated several parameters in men with clinically manifested anogenital HPV infection. The prevalence of specific HPV genotypes in lesions is significant for understanding disease duration, diagnostic approaches, treatment, and follow-up. Knowledge of genotype representation in lesions is crucial for planning national strategies and vaccines. Condylomata, as benign yet highly infectious lesions, impose psychosocial and biological burdens and elevate the risk of acquiring other STIs, including HIV. The treatment of such lesions costs approximately $200 million annually in the USA [47]. Additionally, these lesions can result in significant discomfort, pain, and stigma, affecting individuals’ quality of life and mental health.

### 4.1. Clinical Determinants of Anogenital HPV Infection

Relatively few studies have examined the clinical appearance of condylomata in men, often presenting considerable variation in descriptions due to a lack of universal diagnostic criteria. Diagnosis is typically based on patient history and clinical examination, with biopsies reserved for unclear cases. Molecular assays are not routinely used in men due to the absence of approved kits for external genitalia and limited consensus on indications for their use. Current guidelines discourage molecular assays because of the unclear significance of results in treatment and follow-up. Such inconsistencies highlight the need for unified diagnostic criteria to guide clinical decisions and research efforts.

Our study included men aged 18 to 63 years, with disease duration ranging from two weeks to 20 years (median: 15 months). We identified four condyloma types: fibromatoid, papular, verrucous, and condylomata plana, with verrucous (52.86%) and papular (30.0%) being the most common. Other studies have similarly classified condylomata but with variations in prevalence. For example, a Swedish study reported acuminate (41%) and papular (35%) condylomata in uncircumcised men [48]. The variations observed across studies underscore the need for standardized classification systems to facilitate comparison and improve the understanding of the disease’s clinical manifestations.

### 4.2. Condyloma Localization

In our study, condylomata were most frequently localized on the penis radix (34.39%), shaft (27.14%), pubic region (14.29%), and inner preputial surface (12.86%). These results differ slightly from the Swedish study [49], which found condylomata predominantly on the shaft (33%) and pubic area (29%). We grouped condyloma localizations into four categories: genital (76%), which included condyloma on the radix, shaft, inner prepuce, and coronal sulcus, vs. extragenital (24%), which included condyloma localized just outside of the strict genital region; including pubic, inguinal, scrotal and perianal region; and proximal (41%), including condyloma on the pubic, inguinal, scrotal, perianal region, and the radix, vs. distal (59%), encompassing condyloma on the shaft, inner prepuce, and coronal sulcus, enhancing result clarity. Such categorization provides a framework for further studies to examine patterns of localization and their potential implications for treatment and prognosis.

### 4.3. Molecular Diagnostic Assays

We employed three molecular diagnostic assays—INNO LiPA, HC II, and in-house PCR—on samples from 70 men with clinical signs of anogenital HPV infection. INNO LiPA genotyping assay confirmed HPV presence in 88.6% of the cases, aligning with a Swedish study reporting 92% positivity [47]. However, only 45% of PCR-positive results were also positive with HC II, highlighting the variable sensitivity of these methods. INNO LiPA, despite its high cost, demonstrated superior sensitivity due to its ability to detect small HPV DNA quantities, even in suboptimal samples [50].

Comparison with other studies revealed similar limitations. For example, a Brazilian study reported a 76.5% concordance between HC II and PCR methods [51]. Discrepancies are attributed to intrinsic biases, cross-reactions, and varying technical specifications of assays. While HC II and PCR detect HPV presence, INNO LiPA provides detailed genotype information, underscoring its utility despite its cost and technical demands. Such findings suggest a need for balancing cost and accuracy when selecting diagnostic tools in clinical practice.

### 4.4. HPV Genotypes in Condylomata

Our findings showed that 70% of INNO LiPA-positive samples had one HPV genotype, while 18.6% had two or three genotypes. High-risk (HR) HPV genotypes were confirmed in 20% of the cases, often as part of mixed infections. Similar results were reported in Swedish and multinational studies, where LR genotypes dominated single-type infections, and HR genotypes appeared more frequently in mixed infections [47,52]. The detection of HR genotypes in condylomata emphasizes the need for monitoring and further investigation to understand their potential role in disease progression.

Interestingly, condylomata with mixed infections were predominantly found in the distal genital area. This aligns with studies suggesting that HR genotypes persist longer than LR genotypes, especially in older men [53]. Variations across studies are influenced by population characteristics, sample processing, and geographical factors. Such findings underscore the importance of contextualizing results within specific demographic and geographic settings to inform targeted interventions.

### 4.5. HPV Genotypes and Condyloma Localization

Our results revealed no statistically significant correlation between condyloma type or localization and HPV genotype. However, a greater number of HPV genotypes were detected in condylomata plana and inner prepuce, consistent with findings from Croatian and other international studies [54]. These observations suggest that certain condyloma types and locations may be associated with increased susceptibility to specific HPV genotypes, warranting further exploration to validate these associations.

### 4.6. HPV Negative Results

INNO LiPA-negative cases were consistently negative across detection methods, suggesting true negativity. Negative results may arise from sampling errors, technical limitations, or the presence of HPV genotypes outside the test’s detection scope. Other studies have highlighted similar challenges, emphasizing the need for validated diagnostic methods for male anogenital regions [55]. The identification of false negatives remains a critical issue, as it may lead to underdiagnosis and insufficient treatment, impacting both individual outcomes and public health efforts.

### 4.7. Recent Findings on HPV Carcinogenesis

Beyond its potential impact on somatic, psychosocial, and sexual functioning, as well as the relatively high cumulative cost of treatment and high infection rates, the most significant burden of anogenital HPV infection lies in its oncologic implications. In this context, the increasing contribution of HPV infections to premalignant and malignant lesions in the anogenital region has been observed. In PeIN III lesions, high-risk (HR) HPV genotypes—particularly HPV 16—are confirmed in 75% of the cases, compared to 32% in PeIN I lesions [49]. It is hypothesized that treating high-grade PeIN could reduce the incidence of invasive penile carcinoma, although this has not been definitively proven. While condom use by male partners of women with cervical intraepithelial neoplasia (CIN) has been linked to the regression of CIN lesions, there is still no conclusive evidence confirming that treating HPV-induced lesions reduces the risk of infection and disease progression in women. This, at least in part, explains why routine screening with peniscopy has not become a standard practice [33].

Data on penile carcinoma indicate that the disease may occur in younger men more frequently than previously assumed. Specifically, 19% of the cases are detected in men under 40 years old, while 7% occur in men under 30 years old [56]. In our study, we did not encounter cases of premalignant or malignant lesions. However, given the typically slow progression of the disease, there is usually ample time for intervention when patients seek medical attention promptly. We routinely perform biopsies and histopathological examinations on all atypical or persistent lesions, including therapy-resistant cases. Immunosuppressed patients are examined more frequently, and in cases of atypical lesions, we conduct biopsies earlier and/or at more frequent intervals. For further evaluation, we may also perform HPV DNA typing on biopsy samples to determine the exact HPV genotype(s). This additional testing not only enhances our diagnostic accuracy but also informs our approach to treatment planning and patient monitoring.

### 4.8. Current Analysis and Future Directions

Studies on asymptomatic men dominate HPV research, but they differ significantly from those on clinically manifested infections. Standardized diagnostic methods, indications for HPV genotyping, and guidelines for sampling are urgently needed. Broader, high-quality clinical studies are essential to establish consensus on these issues. Additionally, economic factors should inform the development of new diagnostic methods, ensuring accessibility and affordability. As HPV continues to present significant challenges, prioritizing comprehensive research and innovative solutions is crucial for advancing understanding and management.

## 5. Conclusions

Benign anogenital HPV lesions are not always caused by LR genotypes; HR genotypes were found in 20% of the cases in our study. Mixed infections were present in 15% of condylomata, with no predominant genotype combinations. Commercially available tests like HC II and PCR showed limited sensitivity, detecting HPV in only 45% and 40% of the cases, respectively. INNO LiPA identified HPV DNA in 89% of the cases, emphasizing its diagnostic value.

No statistically significant correlation was observed between clinical manifestations and oncogenic HPV genotypes. However, mixed infections correlated with distal genital localization. An optimal HPV detection method for men is still lacking, necessitating individualized approaches to diagnosis and treatment. As evidence for HPV’s oncogenic potential in men grows, developing standardized, affordable, and sensitive diagnostic methods is imperative for advancing clinical practice and public health initiatives. Continued research and collaboration across disciplines will be vital in addressing the complexities of HPV infection in men, ensuring that advancements in knowledge translate into meaningful improvements in health outcomes.

## Figures and Tables

**Table 1 genes-16-00230-t001:** Distribution of Condyloma Types.

Type	Percentage
Verrucous	52.86%
Papular	30.00%
Fibromatoid	5.71%
Plana	11.43%

**Table 2 genes-16-00230-t002:** HPV Genotype Distribution based on INNO-LiPA analysis.

Genotype	Frequency	Percentage
HPV6	55	69.6%
HPV16	4	5.1%
HPV18	3	3.8%
HPV11	3	3.8%
Others	14	17.7%

**Table 3 genes-16-00230-t003:** HPV Risk Type Distribution based on INNO-LiPA analysis.

Risk Type	Percentage
Low Risk	68.57%
High Risk	5.71%
Mixed (LR + HR)	14.29%

## Data Availability

The original contributions presented in the study are included in the article, further inquiries can be directed to the corresponding author.

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
