# Peer review of "Human Papilloma Virus Infection in Men: A Specific Human Virome or a Specific Pathology?"

_genes, 2025, doi:10.3390/genes16020230_

Round 1

Reviewer 1 Report

Comments and Suggestions for Authors

This submission represents a well-conducted prospective study addressing HPV positivity rates (both high risk & low risk genotypes) in a small male dermatologic cohort. Like the authors correctly point in the Discussion section (lines 246-250), this is an understudied, but most important public health issue “Molecular assays are not routinely used in men due to the absence of approved kits for external genitalia and limited consensus on indications for their use. Current guidelines discourage molecular assays because of the unclear significance of results in treatment and follow-up”.

The word “microbiome” in the title is quite eye-catching, however potentially misleading, as the authors only focus on HPV’s. The authors might consider instead “human virome”.

Patients and Methods.

Based on EU data, almost 35% of 15-year old boys in Croatia have received at least one dose of the HPV vaccine. Since the median age of this cohort was only 30.5yrs, are there any data on how many of the recruited individuals had been previously HPV vaccinated?

It is difficult to fully justify the use of three different molecular assays in two different (paired?) skin specimens, especially for a moderate size cohort (70 patients) and a diverse pathology, in which distinct histopathologic entities might lie adjacently.

  1. The two distinct skin specimens per definition might harbor different genotypes in cases of mixed infections, even for those cases in which one large specimen was divided in two distinct specimens (lines 123-4).
  2. Comparing assays offering full genotyping with assays offering hr genotyping is questionable for pathological entities predominantly caused by low risk HPV’s. However, individual genotyping and not limited pooled genotyping is in the correct direction.
  3. InnoLipa is not a novel assay; on which grounds was it selected over other assays offering extended genotyping?

Discussion

Lines 228-232 and 246-250: These remarks highlighted by the authors indeed represent public health priorities.

4.3. Molecular Diagnostic Assays: The favorable Innolipa clinical performance is well established.

4.7. Recent findings on carcinogenesis: This paragraph needs expanding. Probably this study’s core point is that several HPV skin infections are mixed (both low risk & high risk), translating in increased likelihood for precancers; these considerations are not sufficiently developed, despite being repeated in the Conclusion section.

Conclusion: Comprehensive and concise.

Author Response

Comment 1:

The word “microbiome” in the title is quite eye-catching, however potentially misleading, as the authors only focus on HPV’s. The authors might consider instead “human virome”.

Response 1:

We changed the title as recommended.

Comment 2:

Based on EU data, almost 35% of 15-year old boys in Croatia have received at least one dose of the HPV vaccine. Since the median age of this cohort was only 30.5yrs, are there any data on how many of the recruited individuals had been previously HPV vaccinated?

Response 2:

Unfortunately, we don't have this information.

Comment 3:

It is difficult to fully justify the use of three different molecular assays in two different (paired?) skin specimens, especially for a moderate size cohort (70 patients) and a diverse pathology, in which distinct histopathologic entities might lie adjacently.

  1. The two distinct skin specimens per definition might harbor different genotypes in cases of mixed infections, even for those cases in which one large specimen was divided in two distinct specimens (lines 123-4).
  2. Comparing assays offering full genotyping with assays offering hr genotyping is questionable for pathological entities predominantly caused by low risk HPV’s. However, individual genotyping and not limited pooled genotyping is in the correct direction.
  3. InnoLipa is not a novel assay; on which grounds was it selected over other assays offering extended genotyping?

Response 3:

The results of our work stem from a broader study conducted for the purpose of developing a doctoral dissertation. Consequently, our approach was more scientifically oriented, starting from the premise that we could address questions arising from everyday clinical practice. Unlike in women, there is no clear protocol for screening and diagnosing anogenital HPV infection in men, even though it is very common in clinical practice (being the most common viral sexually transmitted infection). Therefore, we employed the methods available to us at that time which were considered the “gold standard” in diagnosing anogenital HPV infection in women: one detection method (HC II) and one genotyping method (INNO LiPA).

During the study, we noted a relatively high number of negative findings using the HC II method. For this reason, and given our available budget, we included another detection method that was accessible to us—a custom in-house PCR approach.

Comment 4:

4.7. Recent findings on carcinogenesis: This paragraph needs expanding. Probably this study’s core point is that several HPV skin infections are mixed (both low risk & high risk), translating in increased likelihood for precancers; these considerations are not sufficiently developed, despite being repeated in the Conclusion section.

Response 4:

Section is completely rewritten and expanded as requested.

Reviewer 2 Report

Comments and Suggestions for Authors

The current title of the manuscript “Human Papilloma Virus Infection in Men: A Specific Microbiome or a Specific Pathology?” could be modified focusing on findings, for example:

  • Does Human Papilloma Virus Infection Causing Specific Pathology in Men Depends on Virus Type?
  • Sensitivity of Diagnostic Methods, Lesion Variability and Genotype Prevalence of Human Papilloma Virus Infection in Men.

The manuscript is focused on the diagnostic methods, lession variability and prevalence of genotype of Human Papilloma Virus (HPV) infection in men who have sex with men (MSM). The study conducted at the University Hospital Center in Zagreb included 70 men aged 18–65 years with clinically diagnosed anogenital HPV infection is designed and described correctly. After lesions and manifestations of infection were characterized morphologically by two dermatovenereologists, all lesional skin or mucosa samples collected from each patient were studied using different molecular methods. Firstly genotyping of HPV was carried out with INNO-LiPA, a reverse hybridization assay capable of identifying 32 HPV genotypes. Later presense of HPV was examined through either Hybrid Capture II (HC2) or by in-house PCR assay method based on amplification of 450 bp fragment of the HPV L1 region which can detect over 50 HPV genotypes. Authors used multivariate statistical analyses to determine associations between clinical parameters and HPV genotypes.

The findings of the two HPV detection methods were compared with INNO-LiPA genotyping results. Authors indicated that “among 40 samples tested with the HC II assay, high-risk (HR) HPV DNA was detected in 45% of cases. The PCR assay conducted on 50 samples revealed 40% positivity, while INNO-LiPA was positive in 62 of 70 (88.6%) samples”. It is not clear why authors omited some part of samples and do not carried on HC II assay and PCR assay on all 70 specimens?

A diagnostic method INNO-LiPA should be added to the title of Table 2. and Table 3., for exaple: “HPV Genotype Distribution based on INNO-LiPA analysis” and “HPV Risk Type Distribution based on INNO-LiPA analysis”.

Lines 199-200: I am not sure if the sentence “The INNO-LiPA method demonstrated its superiority in identifying HPV genotypes detecting a broader range and higher frequency of positive results compared to HC II and PCR” is inserted in the right place of the manuscript. I suggest this sentence could be inserted between lines 168 and 169.

Discussion and conclusions is well structured revealing clinical and molecular features of condylomata and their association with HPV genotypes and crucial role of man in disease transmission and the need for inclusion of man diagnostics in development of preventive strategies.

Author Response

Comment 1:

The current title of the manuscript “Human Papilloma Virus Infection in Men: A Specific Microbiome or a Specific Pathology?” could be modified focusing on findings, for example:

  • Does Human Papilloma Virus Infection Causing Specific Pathology in Men Depends on Virus Type?
  • Sensitivity of Diagnostic Methods, Lesion Variability and Genotype Prevalence of Human Papilloma Virus Infection in Men.

Response 1:

Thank you for the suggestion. The other reviewer had the similar suggestion regarding the title of the manuscript so we changed it accordingly.

Comment 2:

The findings of the two HPV detection methods were compared with INNO-LiPA genotyping results. Authors indicated that “among 40 samples tested with the HC II assay, high-risk (HR) HPV DNA was detected in 45% of cases. The PCR assay conducted on 50 samples revealed 40% positivity, while INNO-LiPA was positive in 62 of 70 (88.6%) samples”. It is not clear why authors omited some part of samples and do not carried on HC II assay and PCR assay on all 70 specimens?

Response 2:

The reasons for this are purely technical. In some cases, the amount of sample collected from patients was not sufficient to apply all methods. Consequently, we were unable to analyze all 70 samples using every method. However, because the choice of which methods were applied to the samples was essentially random, we consider that the samples not analyzed by a particular method are missing completely at random (MCAR). Therefore, we regard our conclusions as statistically valid.

Comment 3:

A diagnostic method INNO-LiPA should be added to the title of Table 2. and Table 3., for exaple: “HPV Genotype Distribution based on INNO-LiPA analysis” and “HPV Risk Type Distribution based on INNO-LiPA analysis”.

Response 3:

Thank you, good suggestion. Changed accordingly.

Comment 4:

Lines 199-200: I am not sure if the sentence “The INNO-LiPA method demonstrated its superiority in identifying HPV genotypes detecting a broader range and higher frequency of positive results compared to HC II and PCR” is inserted in the right place of the manuscript. I suggest this sentence could be inserted between lines 168 and 169.

Response 4:

Thank you for the suggestion. Changed accordingly.